# Universally Converging Representations of Matter Across Scientific Foundation Models

**Sathya Edamadaka**[†]
MIT
sathyae@mit.edu

**Soojung Yang**[†*]
MIT
soojungy@mit.edu

**Rafael Gómez-Bombarelli**[*]
MIT
rafagb@mit.edu

## Abstract

Scientific foundation models are rapidly emerging across physics, chemistry, and biology, yet it remains unclear whether they converge toward a shared representation of matter or remain governed by domain and modality. We analyze embeddings from nearly 60 models spanning molecules, materials, and proteins, using two complementary alignment metrics to probe their learned representations. We find modest cross-modality alignment for molecules and materials but strong alignment among protein models. We find that training dataset, rather than architecture, is the dominant factor shaping latent spaces. We see some hint of models converging into an optimal solution for the representation space, as nontrivial cross-modal alignment and strong alignment within modalities indicate. However, models align more strongly out-of-distribution than in-distribution, suggesting they remain data-limited and fall short of true foundation status. Our framework establishes representation alignment as a dynamic criterion for evaluating foundation-level generality in scientific models. This is an abbreviated, work-in-progress submission of our full manuscript, which will be linked in the comments below shortly.

## 1   Introduction

The success of foundation models in language and vision has accelerated a parallel effort in the sciences, where large-scale models are trained directly on experimental and simulated data to learn the behavior of physical systems. These machine learning models take as input different descriptions of matter and output predictions of their properties, whether operating on 3D atomistic coordinates to predict energies and forces (e.g., MACE-OFF Kovács et al. [2025], ESEN Fu et al. [2025]), or protein sequences to predict folded structures (e.g., AlphaFold Abramson et al. [2024]). By scaling up model sizes and training on large, diverse datasets, these models learn compact latent representations of the natural world that support generalization far beyond the data seen during training.

There is growing evidence that representations universally converge in language and vision models. For example, the Platonic Representation Hypothesis [Huh et al., 2024] shows that as vision and language models improve, their learned representations converge and become increasingly similar. Another study showed that embeddings from models with different architectures, training data, and model sizes can be translated to and from each other without encoders or paired data [Jha et al., 2025]. In our preliminary work, we have found hints of such universality in biology: Yang et al. [2024] reported strong alignment between protein language models and protein structure models, including shared global features such as intrinsic dimensionality.

This raises a natural question across chemistry, materials science, and biology: are scientific foundation models converging to a universal representation of matter? Just as scientists believe that all matter obeys a set of physical laws, there may exist a corresponding statistical model of the joint

---

[†]Equal contributions
[*]Corresponding authors

39th Conference on Neural Information Processing Systems (NeurIPS 2025) Workshop: AI4Mat: AI for Accelerated Materials Design.

distribution over matter. Answering this would accelerate scientific machine learning even outside foundation model development, as researchers are already leveraging representations from one model to improve performance in others of different modalities and domains Yu et al. [2024]. As an example, representations from atomistic machine learning interactive potentials (MLIPs) pre-trained on small molecules were used to accelerate Boltzmann emulator training Pinede et al. [2025] and predict NMR chemical shift of proteins Bojan et al. [2025].

We use latent space similarity metrics as a proxy for representational alignment, and find that scientific foundation models across different modalities, training tasks, and architectures are strongly aligned. We further show that representations converge toward a shared universal representation of matter as model performance improves. We then establish representational similarity as a dynamic benchmark for foundation-level generality by probing embeddings of both in-distribution and out-of-distribution structures. Our analysis covers 52 models across multiple input modalities (SMILES, 3D atomistic coordinates, protein sequences, protein structures, and natural language), architectures (equivariant and non-equivariant MLIPs, conservative vs. direct prediction models), and training domains (molecules, materials, and proteins) spanning five datasets. We use three alignment metrics that vary in locality and symmetry. Further details on our analysis are available in appendix section B. This submission is a highly abridged rendition of our full analysis, which will be available shortly.

## 2 Representations of matter in scientific foundation models are converging

We first examine whether models trained on different input modalities have similar latent representations. We then show how alignment increases with potential energy prediction accuracy, thus indicating convergence to an optimal representation of matter and supporting the Platonic Representation Hypothesis Huh et al. [2024]. To test whether similar convergence arises in chemical domains, we used the multi-modal QM9 dataset, which provides SMILES strings, SELFIES strings, 3D atom coordinates, and total potential energy labels (calculated with density functional theory, or DFT) of nearly 134,000 small molecules. By extracting embeddings of 50,000 of these chemically simplistic structures from each model and calculating the pairwise correlation between each set of embeddings, we demonstrate how correlated different models' representations are.

### 2.1 Models are strongly aligned within and across modalities

Fig. 1A shows the alignment between each class of models, generated by averaging the full representation correlation matrix (Fig. 2) across models of the same architecture. Models of the same modality are strongly aligned, as seen by the dark off-diagonal triangle in Fig. 1A. Specifically, models of vastly different architectures that take in 3D atomistic coordinates are all well-aligned. The same trend is true for models that operate on SMILES or SELFIES string encodings. We also observe alignment between SMILES-based models (Molformer, ChemBERTa) and atomistic MLIPs, with particularly strong alignment between text-based models and Orb architectures. Although their CKNNA values may seem low compared to those for within-modality alignment, they exceed the highest CKNNA values previously found between language and vision foundation models Huh et al. [2024]. Although representations derived from string encoding lack conformational geometry, the string encodings contain information that is effectively equivalent to molecular graphs. Because the conformers in QM9 have minimal structural variance around their lowest-energy conformers, the molecular graphs would capture most of the informational content relevant to the QM9 dataset, which explains the nontrivial alignment between string-based and 3D coordinate-based models. In addition, large natural language models (LLMs) such as DeepSeek and GPT align strongly with string encoding-based materials models and show similar alignment scores with OrbV2 as smiles-based models. Protein models show even stronger cross-modality convergence. Reproducing the protein alignment analysis of Yang et al. [2024], but using CKNNA instead of IDCor, we find that embeddings from protein sequence models (e.g., ESM2) and protein structure models (e.g., ProteinMPNN) align as strongly as the best molecular cases, suggesting greater universality in the protein domain (Fig. 3).

### 2.2 Alignment increases with model performance

Vastly different scientific models aren't just highly aligned; they are converging. We show in Fig. 4 that as models improve at predicting the total energy of inputted structures, they also grow more aligned with the best-performing model. We illustrate this using subsampled OMat24 as a benchmark. Further information on this is available in appendix section B.6.

## 2.3 Local and global alignments follow similar trends

CKNNA is highly sensitive to changes in local neighborhood structure. Therefore, in principle, there can be cases where CKNNA is low even when global representation space structure is similar, or vice versa. To assess whether such cases occur here, we characterize global structure using two metrics. Intrinsic Dimension estimates the minimal number of variables needed to describe the data manifold, and Intrinsic Dimension Correlation (IDCor) quantifies mutual information shared between two data manifolds. High IDCor but low CKNNA could imply globally similar topology but warped geometry that disrupts local neighborhoods (e.g., a straight line versus a highly warped sinuous curve). Conversely, high CKNNA but low IDCor could imply locally similar neighborhoods but globally different density distribution and clustering.

In our results (Figure 5), we do not observe either edge cases. Increasing $k$, the neighborhood size, in CKNNA reduces local sensitivity of the metric, yet the qualitative trends are consistent across both ID/OOD datasets and across models. CKNNA and IDCor consistently co-vary. Together, these results indicate that for the chemical foundation models, the local similarity is reflective of global similarity and vice versa. This is further shown in Figure 6 with information imbalance, further discussed in section 3.2.

## 2.4 Consistent intrinsic dimensionality across models suggest statistical convergence of representations

Intrinsic dimensions are dataset-dependent, and can be interpreted as an estimate for the intrinsic complexity of information encoded in the latent space. Across invariant features, ID varies with the dataset but is mostly consistent across models within each dataset: QM9 representations have relatively low intrinsic dimensionality ($\sim 5$), whereas OMat24 ($\sim 10$), sAlex ($\sim 8$), and OMol25 ($\sim 10$) have higher intrinsic dimensionality, as shown in Figures 7—10. This likely reflects differences in the diversity of chemical environments and conformations sampled in each dataset. Even though QM9 and OMol25 both contain predominantly organic molecules, QM9 primarily consists of near-equilibrium, low-energy conformers, whereas OMol25 contains more non-equilibrium and higher-energy conformers and therefore spans a broader variety of chemical environments.

Importantly, intrinsic dimension values fall within a relatively narrow band for each dataset (e.g., 7 to 10 on OMat24), consistent with the idea that matter representations of different models share a relatively universal low-dimensional structure. The major outliers are the equivariant features of MACE MP (large/medium) models and the Orb models (neither invariant nor equivariant), which show higher IDs (e.g., 17 to 20 on OMat24). This is expected, as these features encode explicit rotational degrees of freedom, in addition to invariant features. The same pattern holds across all datasets.

## 2.5 Alignment can emerge between non-equivariant and equivariant models via equivariance-inducing training

Different MLIPs make different design choices, particularly in roto-invariance/equivariance and conservatism. While conservative models obtain forces via taking gradients of energy with respect to atomistic positions, direct models predict forces directly. Roto-invariant/equivariant models by construction guarantees forces to rotate with the structures and energies to be invariant with rotations. Respecting these symmetries in architecture generally increases compute cost, but they have been known to improve sample efficiency in training and model performances. However, recent results such as Orb models have challenged this tradeoff, achieving good model performance while being non-equivariant and non-conservative.

Among all models and datasets, the OrbV3 conservative model consistently shows the highest average alignment with others, and is also among the strongest performing models. This is notable because OrbV3 is not architecturally roto-invariant/equivariant. This nontrivial, strong alignments can be attributed to OrbV3's newly introduced roto-equivariance-inducing regularization scheme, *equigrad*. By regularizing the gradient of energy with respect to identity rotation matrices, the model learns energy quasi-invariance and force quasi-equivariance, the same inductive biases that are built directly into models like MACE, but at much lower inference and training costs. In contrast, non-conservative OrbV3-direct model, which was not trained with *equigrad*, show weaker alignment with other equivariant models.

This illustrates our work's utility as a method to validate architectural decisions from only inputting unlabeled structures. By gauging alignment between cheaper and expensive models for any desired input structure, the cheapest model for a desired level of representational power can be chosen.

## 3 Materials models remain data-governed and not yet fully foundational

### 3.1 Model phylogenetic tree reveals dominant effect of training dataset in shaping representation space

To visualize model similarity, we constructed an evolutionary tree of scientific foundation models using CKNNA-derived distances (Fig. 11), where longer branches indicate greater differences in alignment. The tree recovers expected clusters by architecture, dataset, and model size: models trained on similar small-molecule datasets—such as string-based models, Geom2Vec, and MACE-OFF23—cluster together, whereas MACE-MP aligns with other MLIPs trained on the materials (MP) dataset, underscoring that training data often shapes representation geometry more strongly than architecture. When dataset effects are controlled, models with similar architectures cluster more closely, with equivariant models (Eqv2, eSEN, UMA, MACE) forming one group and invariant ones (Orb V2, V3) another. Interestingly, large language models (DeepSeek, GPT, Qwen3) group with chemical language models (ChemGPT, MolFormer) despite their modality differences, suggesting that LLMs encode chemical and compositional features akin to molecular language models. This alignment hints at a representational bridge between LLMs and chemical models, motivating future work on how prompting and modality interplay shape cross-domain chemical understanding.

### 3.2 Models show strikingly different behavior in ID and OOD

We observed that stronger energy regression performance correlates with greater representational convergence, indicating that models with more transferable latent spaces exhibit higher alignment, while weaker models show lower correlation Huh et al. [2024]. In other words, foundation models are more aligned on in-distribution (ID) inputs than on out-of-distribution (OOD) inputs. Following from training dataset effects from the previous section, how does alignment change when models are evaluated on data within their training distribution compared to data that is relatively out-of-distribution?

For in-distribution analysis, we sampled 50,000 embeddings from sAlex and OMat24, both overlapping with many models' training data, and used OMol25 as the OOD benchmark. In-distribution results (Fig. 12) show clear dataset effects: models trained on the same dataset align more closely, even across architectures, while those trained on different datasets diverge. For instance, eSEN trained on OMat24 aligns more with EqV2 OMat models than with eSEN trained on MPTraj, and the near-identical OMat24 and sAlex CKNNA matrices reflect their shared structures. Out-of-distribution results show the opposite trend. Alignment scores increase overall, and clustering shifts from dataset-to architecture-driven, suggesting that when tested on unseen data, models collapse to noisier, less informative representations that inflate apparent similarity. This behavior highlights that current materials models remain data-limited and have not yet achieved foundation-level generality despite scaling improvements. Unlike measures of latent space complexity or local neighborhood similarity, information imbalance (II) explicitly quantifies whether one representation contains more information about another, and in which direction, particularly useful for probing how models share or diverge in their learned representations. For out-of-distribution structures from OMol25 (Fig. 13), nearly all models appear to capture very similar information, especially when compared to OMat24. This particular convergence is unlikely to reflect genuine universality and instead suggests that models are either producing similar noise in under-trained regions of latent space or able to capture only trivial information about the inputted structures common to all embeddings.

## 4 Conclusion

We find strong statistical convergence in learned representations of matter across 50 models, 5 datasets, and 4 metrics. Models trained on diverse modalities and architectures exhibit substantial alignment, exceeding what was seen between language and vision models, with higher-performing MLIPs seemingly converging toward a shared solution manifold. Moreover, equivariance-inducing training can yield emergent alignment even for non-equivariant architectures, indicating that similar inductive biases can be learned in latent representation space. However, these models are still largely data-governed, and not yet foundational. Our framework establishes representation alignment as a dynamic benchmark for evaluating foundation-level generality and representational power in scientific models.

## Acknowledgments and Disclosure of Funding

We thank Juno Nam, Jinyeop Song, Lucas Pinede, Matteo Carli for helpful discussions. S.Y. thanks Ilju Foundation for PhD fellowship support.

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

## A  Supporting figures

## B  Analysis Details

### B.1  Models

Scientific foundation models in the materials space fall into several families. The first is Machine-learning Interatomic Potentials (MLIPs), which operate on 3D atomic graphs and are trained to predict energies/forces (often with a conservative objectives that tie forces to the energy gradients). MLIPs can have equivariant architectures, such as MACE (many-body message passing with spherical

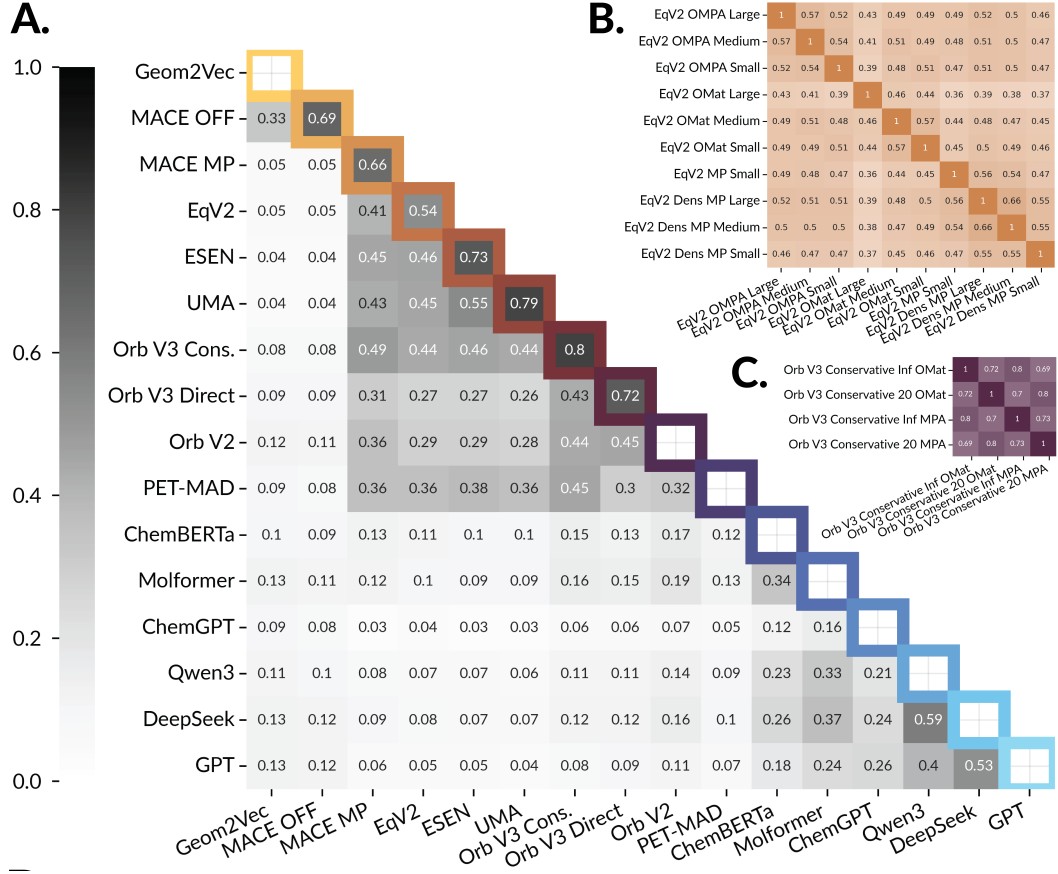

Figure 1: QM9 CKNNA matrix. (A) shows a block-diagonalized CKNNA correlation matrix across models. Each row corresponds to models grouped by architecture, and every square represents a block-diagonal-averaged portion of the full CKNNA matrix shown in Fig. 2. As a result, some elements along the diagonal are not 1, as they are averages of the CKNNAs of all models within that class. Two representative block matrices with multiple, averaged within each architecture family, are shown for EquiformerV2 (B) and Orb V3 Conservative (C) to illustrate this. The rows that have a single model in the architecture family, e.g. Geom2Vec, Orb V2, and DeepSeek, trivially have a CKNNA of one in the diagonal and thus has been removed to emphasize the rows that represent entire groups of model embeddings.

harmonics) [Batatia et al., 2023], EquiformerV2 (i.e. EqV2, an equivariant transformer that scales to higher-degree tensor reps) [Barroso-Luque et al., 2024], and eSEN (an equivariant energy network emphasizing smooth, stable potential energy surfaces for property prediction) [Fu et al., 2025]. In an effort to pretrain a single model on as much atomistic data as possible, UMA ("Universal Models for Atoms") [Wood et al., 2025] is trained jointly across molecules and materials to enable cross-domain transfer on energy, force, and property prediction tasks. There are also non-equivariant MLIPs that can be trained either directly (outputting forces only) or conservatively (outputting energy and forces); the Orb [Qiao et al., 2022] family explicitly provides both regimes. We also include a lighter-weight representation encoder, Geom2Vec [Pengmei et al., 2025], which maps molecular geometry into representations without force-field training, serving as a lower-performance 3D atomistic model to test the Platonic Representation Hypothesis. Last are text/sequence encoders and decoders over string representations: ChemBERTa, a RoBERTa-style masked-language model on SMILES) [Ahmad et al., 2022], MolFormer (a transformer-based SMILES chemical language model) [Ross et al., 2022], and ChemGPT (a decoder-only netotypically SELFIES/SMILES, generative) [Frey et al., 2023]. In addition, we also included three open-sourced natural language models: GPT OSS 20B [OpenAI et al., 2025], Qwen3 30B A3B Thinking [Yang et al., 2025], and DeepSeek R1 distilled onto Llama 8B [DeepSeek-AI et al., 2025]. As they're all trained on extensive, several-trillion-long token

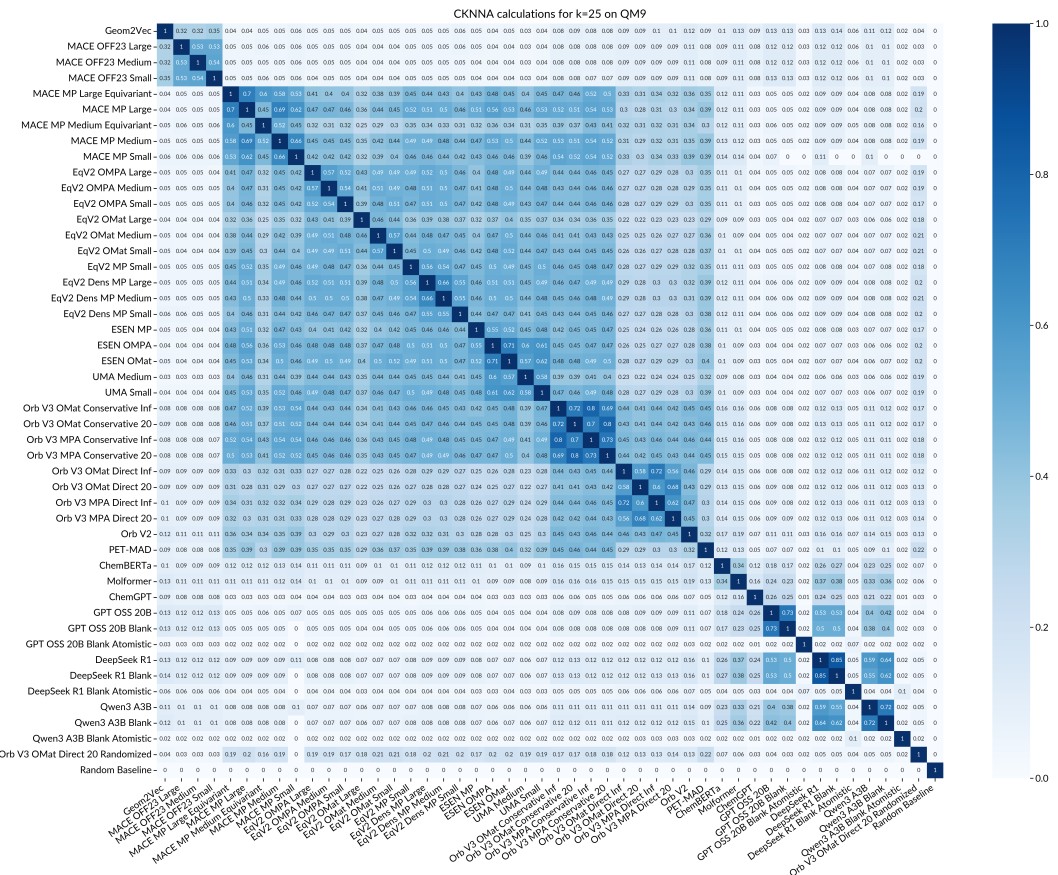

Figure 2: Full CKNNA matrix computed with QM9 data.

datasets, they should encode general chemical knowledge and serve as additional language models trained on different training tasks.

## B.2 Datasets

QM9 [Butler et al., 2018] is a benchmark dataset of 134k, small organic molecules with DFT-computed properties. sAlex is a subsampled version of the Alexandria dataset (10M inorganic materials structures), curated for diversity and reduced redundancy. OMat24 [Barroso-Luque et al., 2024] or Open Materials 2024 is a massive dataset (118M structures) of inorganic materials, enabling broad coverage across domains. OMat24 was created by applying different transformations to Alexandria structures to yield far-from-equilibrium materials. OMat24 and sAlex were used to train most of the materials models in our analysis, and thus are considered to be in-distribution structures. OMol25 [Levine et al., 2025] or Open Molecules 2025 is a >100M molecule dataset computed at $\omega B97M - V$ level of theory, spanning organic, biomolecular, electrolyte, and transition-metal complexes. This dataset is comprised of large structures and came out after most of the models in our analysis were trained, and thus serves as an out-of-distribution dataset. RCSB, or the Protein Data Bank (PDB), comprises >200k experimentally determined protein structures. By filtering for structures with available corresponding sequences, they were used for representation alignment across sequence-based and structure-based protein foundation models. For each of the first four datasets, we subsample roughly 50,000 random samples for our analysis, and we chose roughly 19,000 for the last dataset given a lack of available, high-fidelity sequence-structure protein pairs.

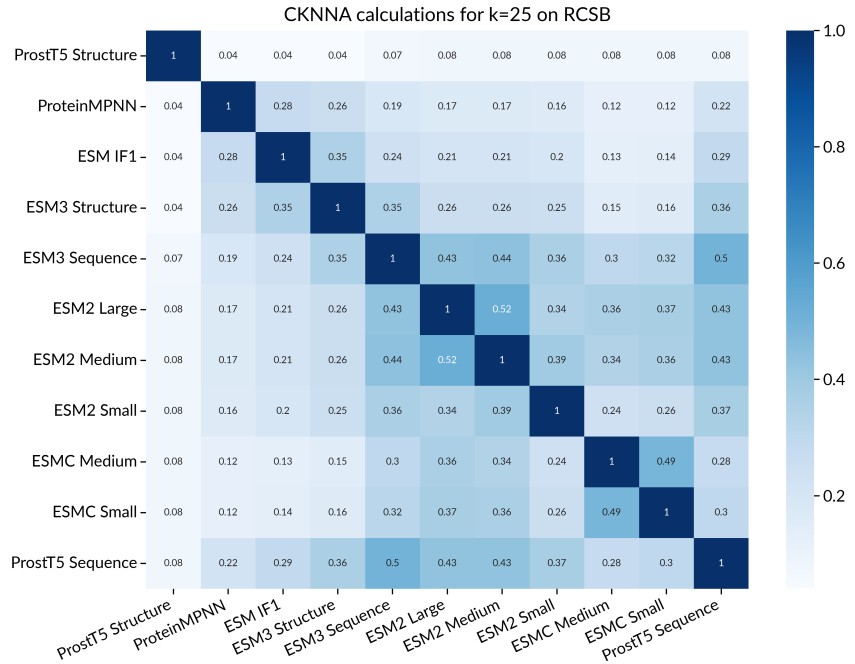

Figure 3: CKNNA matrix of protein models computed with subsampled RCSB proteins.

### B.3 Alignment metrics

We consider three alignment metrics that vary in locality and symmetry. Centered Kernel Alignment (CKA) metric measures how similar is the distance induced by one representation to the distance induced by another. CKNNA [Huh et al., 2024] is a modified CKA measure that computes the kernel alignment only for its nearest neighbors. Intrinsic dimension can be described as the minimum number of degrees of freedom needed to represent the data with minimal information loss. IDCor [Basile et al., 2025] robustly estimates the mutual information shared between the underlying high-dimensional manifolds, even in complex nonlinear relationships. Higher IDCor values indicate greater redundancy between representations. Lastly, information imbalance is an asymmetric, local metric that quantifies which models' latent spaces contain more information than others.

### B.4 Extracting embeddings for analysis

Embeddings were extracted by inputting ASE atom, graph, or smiles string inputs into each model and saving the last hidden layer features from each model. If features were outputted for each node (atom) in the inputted structure, we averaged them across all nodes to get a structure-wide embedding independent of the size of the input. We will release a public repository with all embedding extraction and analysis code shortly at https://github.com/learningmatter-mit. For QM9, we took the lowest-energy conformer from each of the 133,885 structures and generated embeddings for each. For OMat24, OMol25, and sAlex, we randomly sampled 150,000 structures and generated embeddings for each. For RCSB, we used the PISCES PDB culling tool to select proteins with 0-3.0 Åresolution, an R-factor of 0.25, sequence lengths between 40 and 1000, sequence percentage identity of $\leq 70\%$, X-ray entries included, NMR entries included, chains with chain breaks included, and chains with disorder included, thus selecting 48,530 PDB sequence-structure pairs.

### B.5 Analyzing embeddings

We randomly selected 50,000 of the same embeddings per dataset (selecting all 48,530 embeddings for RCSB) to use for calculations, as we were constrained by 192 GB of CPU RAM and CKNNA required matrix multiplying all embeddings together (some of which spanned over $(50,000 \times 2048)$ in dimension) to form an embedding similarity kernel. We then calculated the pairwise CKNNA

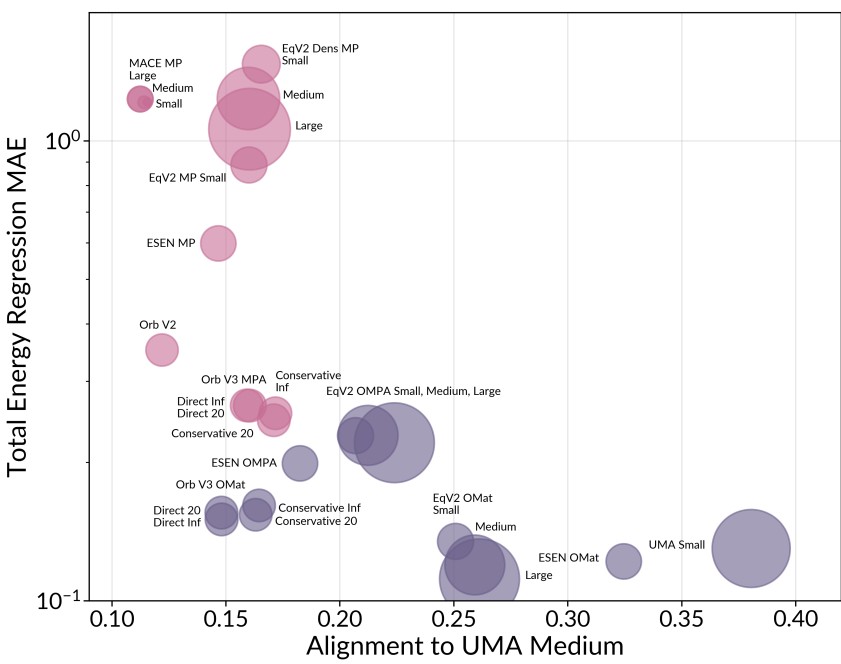

Figure 4: As models increase in Energy MAE regression performance, they improve in alignment to the best performing model. This illustrates the convergence of representations in scientific models, demonstrated on 1000 OMat24 structures.

of each set of embeddings. We then calculated the pairwise CKNNA and IDCor between every pair of model embeddings for each dataset, as well as the intrinsic dimension for each set of model embeddings for each dataset. As the full matrices are unwieldy to show as main figures, we ordered the models in the QM9 matrix together by architecture and averaged them across architectures (e.g. averaging EqV2 small, medium, and large models trained on OMat24, MP, etc. together) to produce Fig. 1.

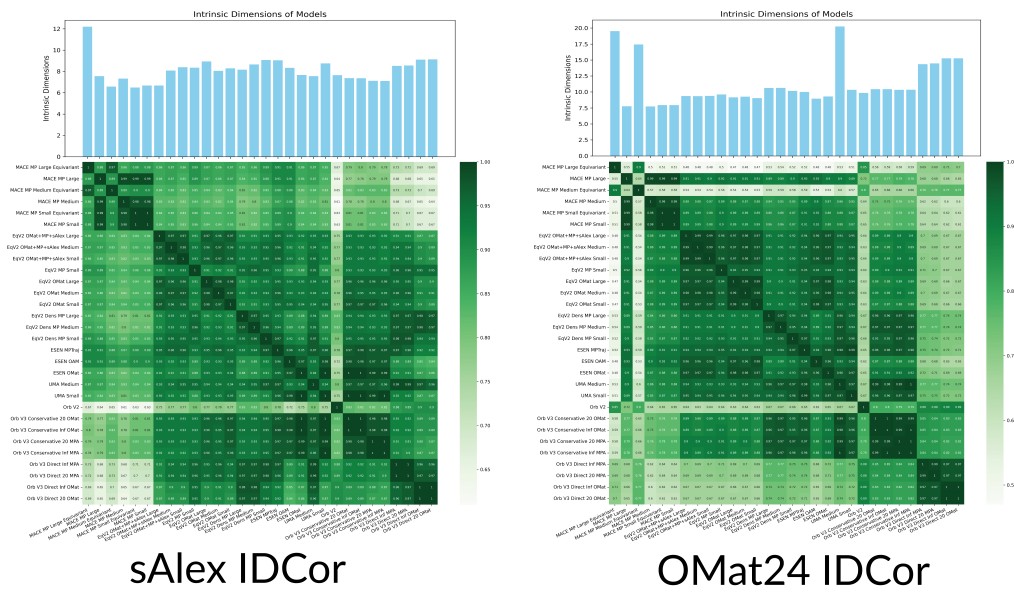

Figure 5: Intrinsic dimension and IDCor matrices of chemical models computed with sAlex and OMat24 data.

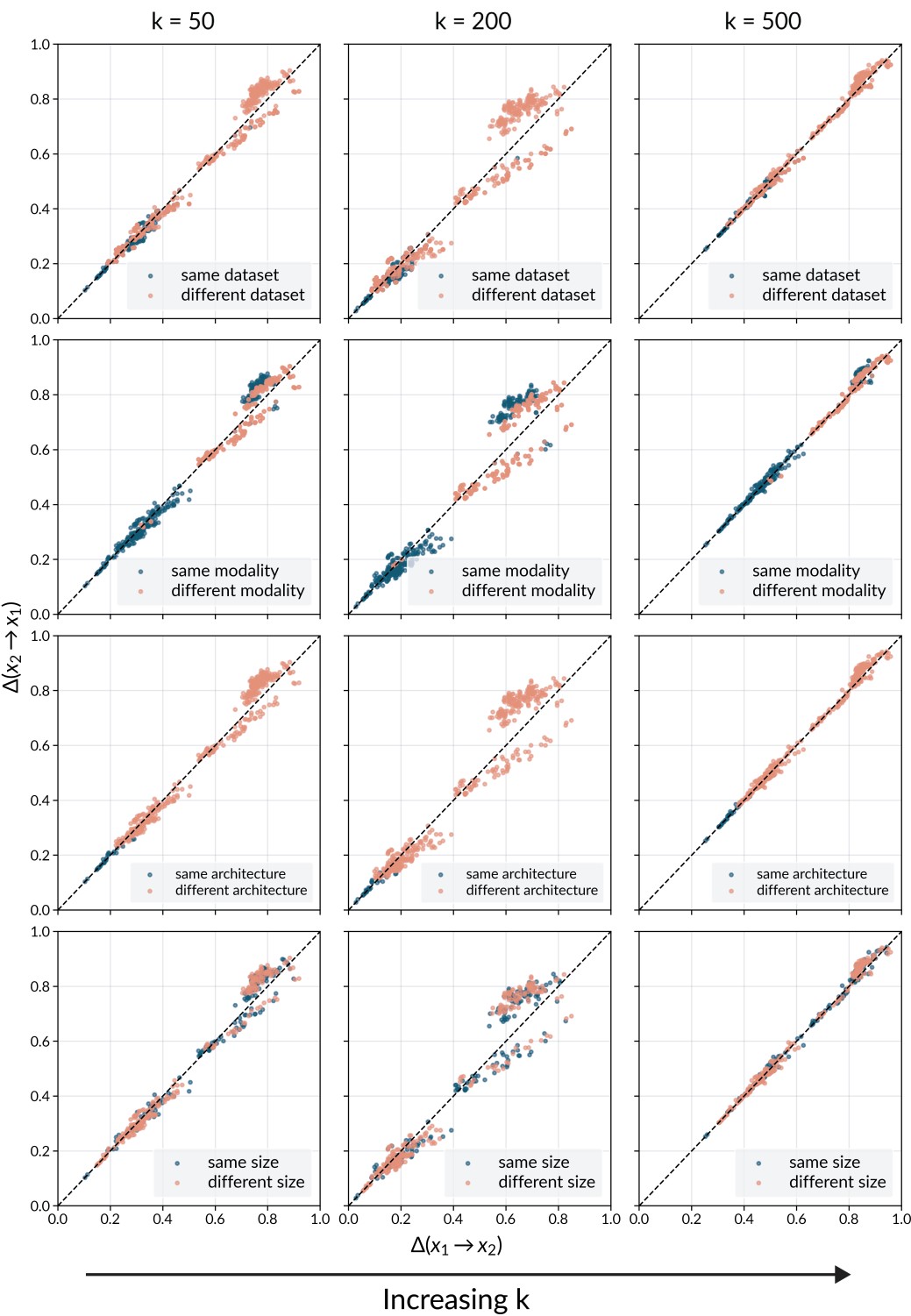

Figure 6: Information imbalance between all pairs of models colored by CKNNA and IDCor. Points in the bottom left indicate nearly identical information between representations, points in the middle indicate shared information, points towards the top right indicate orthogonal information, and off-diagonal points indicate that one representation contains the other's information.

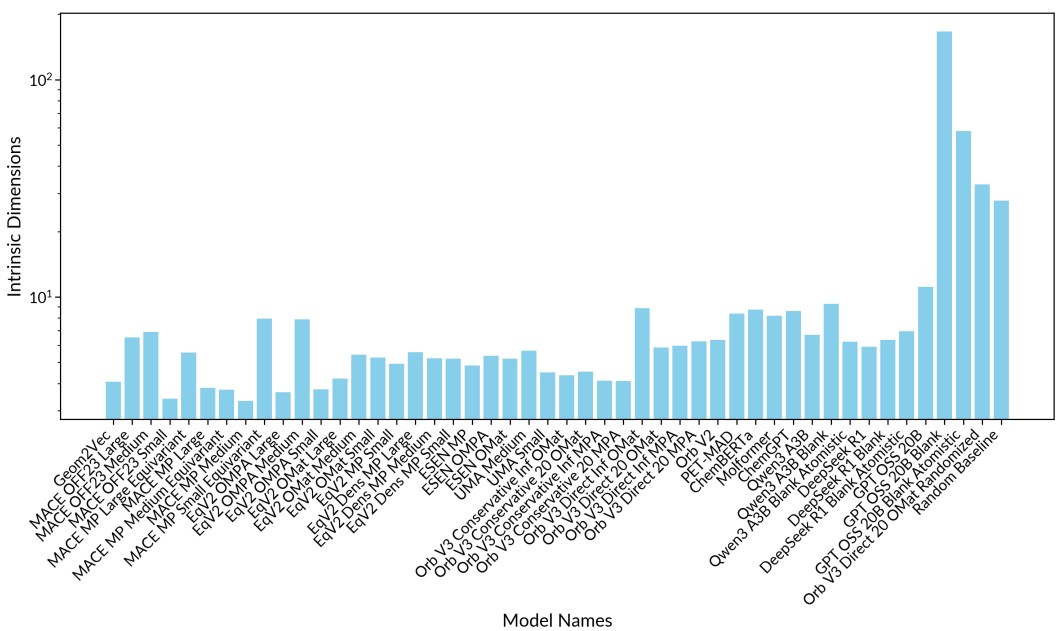

Figure 7: Intrinsic dimension of QM9 embeddings.

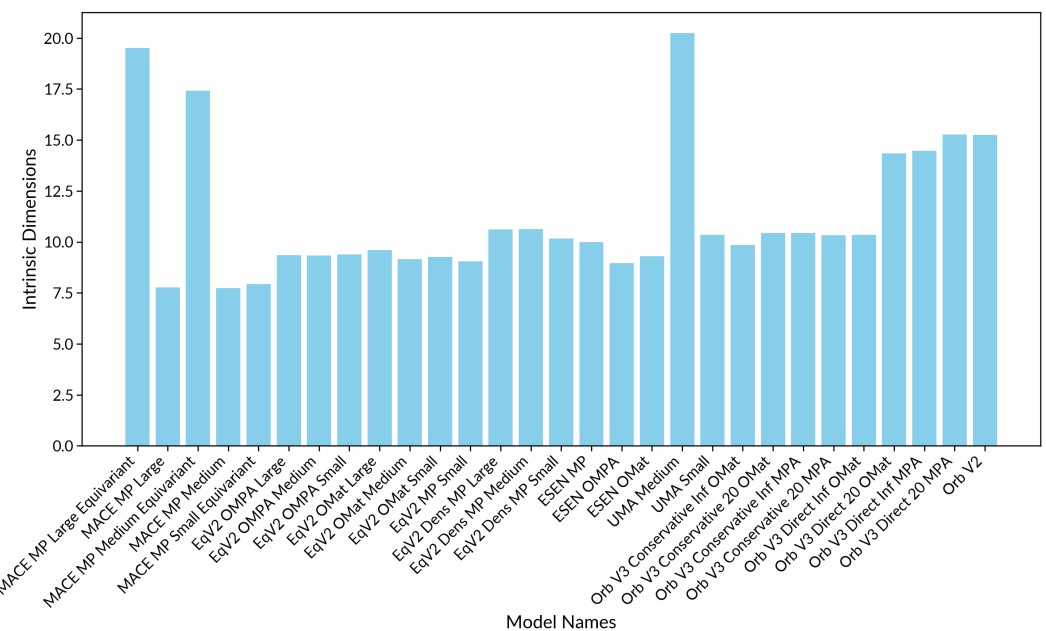

Figure 8: Intrinsic dimension of OMat24 embeddings.

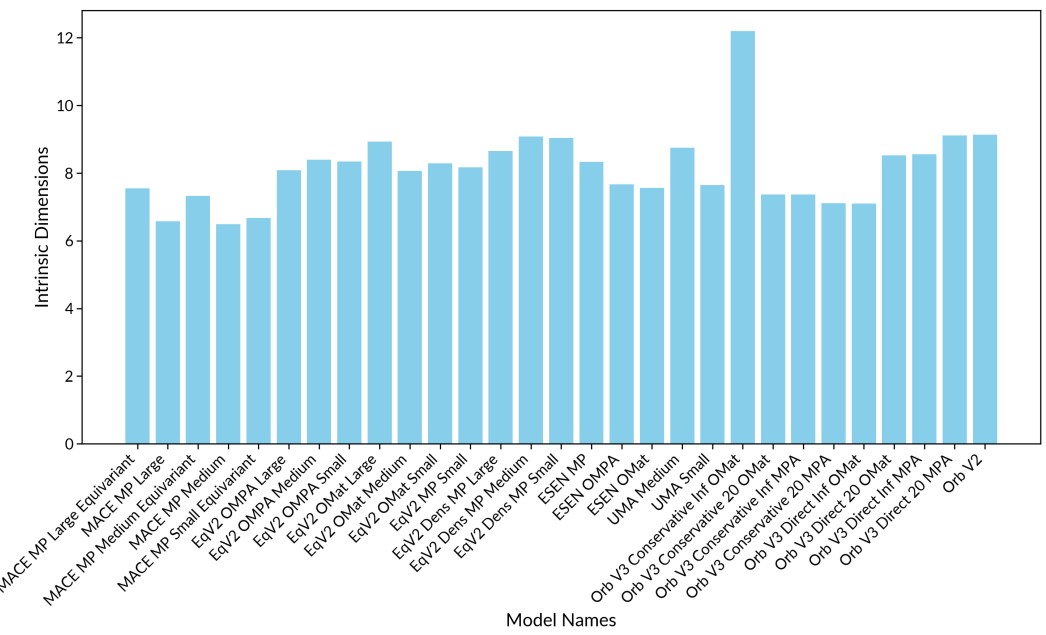

Figure 9: Intrinsic dimension of sAlex embeddings.

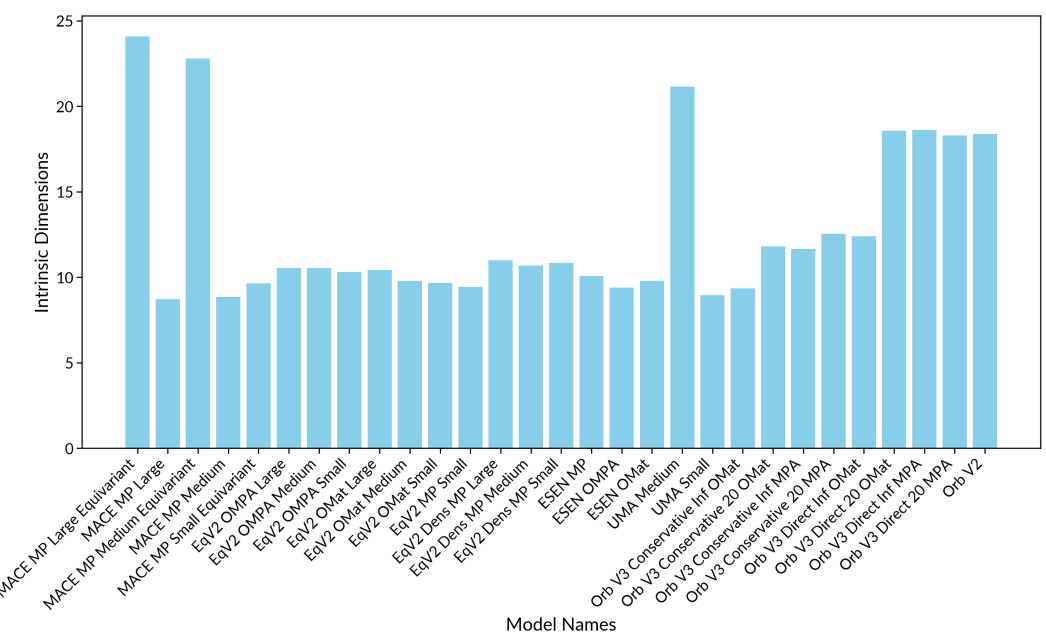

Figure 10: Intrinsic dimension of OMol25 embeddings.

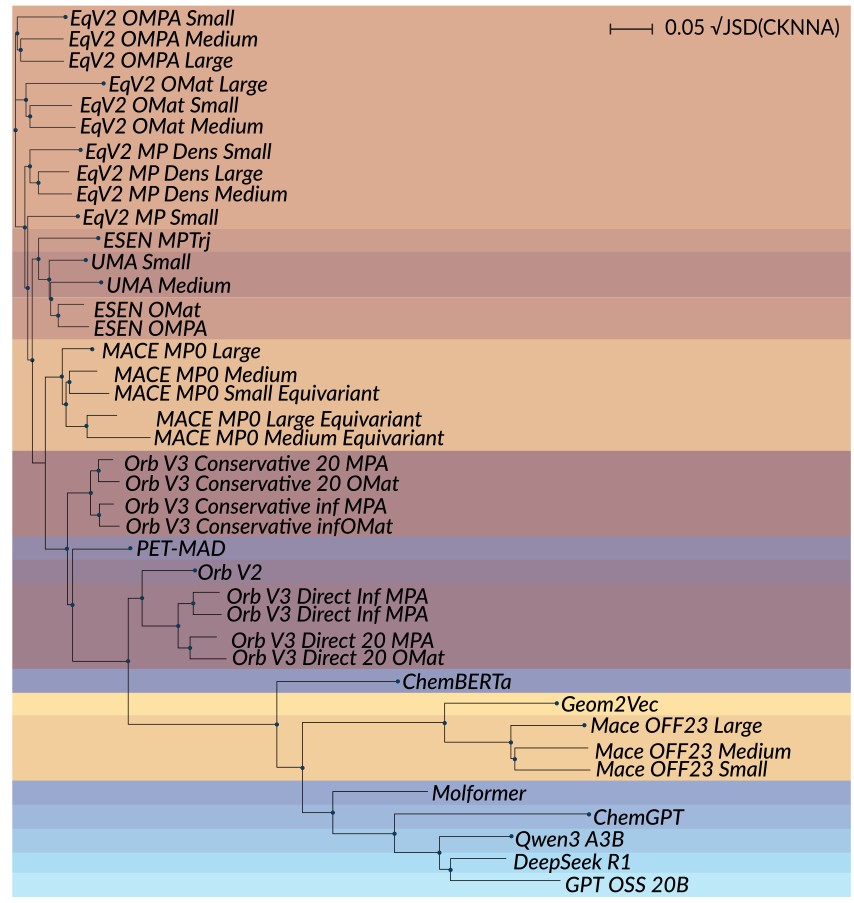

Figure 11: Phylogenetic tree of the models using QM9 CKNNA-derived distances.

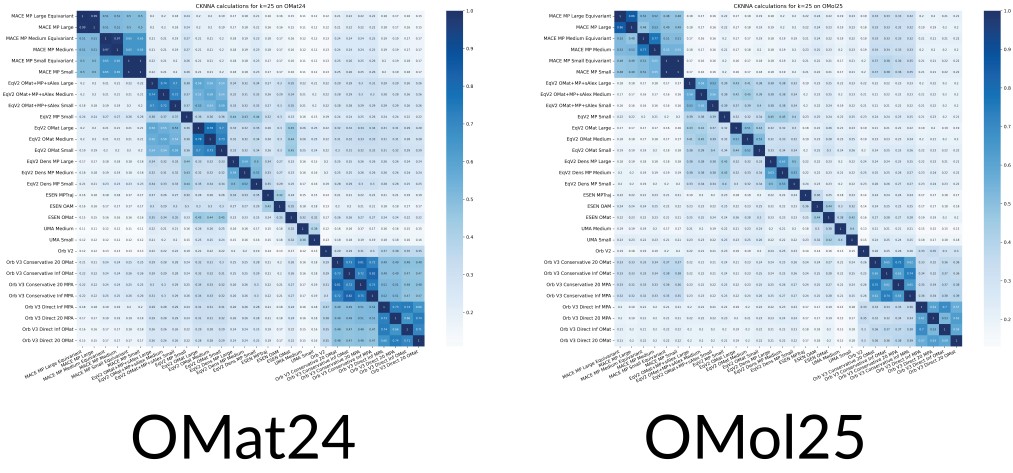

Figure 12: CKNNA matrices of chemical models computed with OMat24 (in-distribution) and OMol25 (out-of-distribution) data.

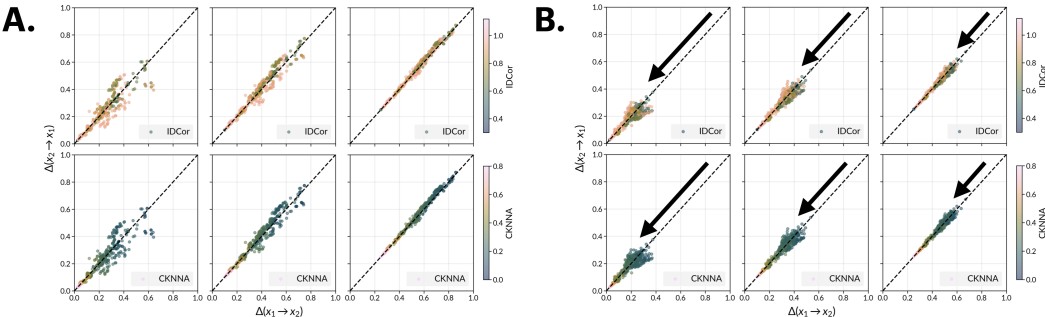

Figure 13: A. Information imbalance for increasing values of k evaluated on OMat24 structures. B. The same measure but evaluated on OMol25 structures. Points towards the bottom left indicate more similar information learned, and points in the upper right indicate more orthogonal/different information learned. Therefore, models are expressing much more similar information, and thus are more aligned, for OMol25 than OMat24, despite OMol25 being completely out of distribution and OMat24 being in-distribution.

