# OpenReview forum: "Universally Converging Representations of Matter Across Scientific Foundation Models"
_NeurIPS.cc/2025/Workshop/UniReps — UniReps2025 oral_

### Official Review · Reviewer_uV1z · 2025-09-05
**Comparing Representations of Matter Across Scientific Foundation Models**

**Confidence:** 4

**Review:**

**Summary:**

This work compares the embeddings of 52 different models using the three similarity measures Centered Kernel Nearest-Neighbor Alignment, Intrinsic Dimension Correlation and Information Imbalance.
The comparisons are done on datasets of various kinds of scientifically interesting matter like small organic molecules, proteins and inorganic materials.


**Strengths:**

**S1:** The work is interesting for further use of foundation models in the natural sciences.


**S2:** There is a nice spread of compared models and similarity measures.


**Weaknesses:**

**W1:** The conclusion of "models converging to a single, optimal representation space" (line 9-10) I think is too strong when
considering the presented results.
This conclusion is also presented in section 4.1 "Significant cross-modality alignment and strong alignments within modalities support convergence".
If we consider figure 1, many of the compared models have similarity of less than 0.15.
Since the maximum is 1, either a good argument should be presented for why this should be considered a high degree of alignment,
or the conclusion should be less strong (see also related questions **Q3**, **Q4**).
Also, the word "significant" should only be used if the results are significant with respect to some statistical test.



**Questions and suggestions:**

**Q1:** It would be nice if you mentioned the "three complementary alignment metrics" (line 6) by name already in the abstract.
When considering similarity and alignment which measure you use is extremely important.


**Q2:** In line 31 you write: "A later work refuted cross-modality alignment" and give Jha et al. 2025 "Harnessing the Universal Geometry of Embeddings"
as a reference. However, in Jha et al. 2025 they only say that it is possible to translate embeddings of different models to the same latent
space. They do not refute anything.


**Q3:** In line 116-117 you write "As reported in prior PRH work, cross-modality alignment scores above 0.16 are already noteworthy".
By noteworthy, do you mean we should consider 0.16 as "similar"? Can you add a reference to the prior work and a short explanation
of why 0.16 is already noteworthy?


**Q4:** Line 151-158: As you note average alignment increases for OOD examples. In figure 7, there are some of the comparisons which when in distribution have 0.16
e.g. ESEN MPTraj, ESEN OAM, but when out of distribution have 0.22 and 0.24.
On the other hand, it seems to me that all the comparisons which have similarities higher than 0.65 go down when switching to the OOD examples.
Does this not challenge the statement about 0.16 from line 116-117?


**Q5:** In figure 1: When you write single model groups, do you mean that you have only one seed of a Geom2Vec model?
If we trained several seeds of the same model type and compared them, would this similarity measure give us something close to one?
And if not, how do we interpret the similarity of this model with models from other families?


**Q6:** Line 177: I think this should reference figure 6.


**Q7:** Line 179: I think this should reference figure 5. And w.r.t. Figure 5: You write that
"higher information imbalances match with lower CKNNA and higher IDCor", but it looks like to me that there are instances of high IDCor for
both low and high information imbalance, but mostly for lower information imbalance.


**Q8:** The caption of figure 6 is quite confusing. I think you need to remove the "colored by CKNNA and IDCor,"?


**Q9:** Appendix B: For reproduction purposes, you should write how you compute the similarity measures you use.

**Score:**

3

**Topic Fit:**

3

---

### Official Review · Reviewer_dsST · 2025-09-10
**Review of Molecular Embedding Alignment Analysis**

**Confidence:** 4

**Review:**

The paper presents a relevant comparison between embeddings of models trained on both organic and inorganic molecules. By employing three different alignment metrics, the authors provide a meaningful analysis of the current state and limitations of commonly used molecular representations.

The paper makes a valuable contribution, as relatively few studies have explored this specific direction. Overall, the quality of the work is good, though certain aspects could be improved, particularly in the presentation of figures and captions.

## Strengths:

- The topic is timely and original, and the comparative study sheds light on underexplored aspects of molecular representations.

- The phylogenetic tree representation in Figure 2 is an intuitive and effective way of visualizing the results.

- The methodological approach is sound, and the analysis provides insights that could inspire further studies.

## Points for Improvement:

- Clarity of Metrics: In Section 3, the authors should explicitly introduce the acronym and reference for the Information Imbalance metric, consistent with how other metrics are described in the appendix.

- Figures and Captions:  Figure 5 lacks sufficient explanation. It is unclear how the plots differ across columns. The accompanying description should clarify this. Additionally, the choice of colormap could be improved: adopting a more distinctive binary variation (e.g., a RdBu colormap) would enhance interpretability. Figures 7 and 8 are too small to be easily interpretable. Increasing their size would improve readability and comprehension.

- Threshold for Alignment Quality: The alignment analysis would benefit from a clearer definition of what constitutes "good" versus "bad" similarity, supported by explicit thresholds or benchmarks.

- Comparison of Organic vs. Inorganic Models: Given the intrinsic differences between organic and inorganic molecules, a deeper comparison of their respective alignments would enrich the study. It is also not entirely clear whether all models were trained on both types of molecules—clarifying this point would strengthen the paper.

- Learned Equivariance: The discussion on learned equivariance is currently the weakest part of the paper. While the authors' insights are interesting, the claims require stronger support. Additional experiments could be designed to test equivariance or invariance, for example by analyzing embeddings of inputs subjected to transformations the model is expected to handle.

## Recommendation:
The paper meets the criteria for acceptance to the workshop. While certain aspects could be refined—particularly the clarity of figures, the discussion on equivariance, and the definition of alignment thresholds—the core idea is strong and the results are promising. With minor revisions, the paper could be substantially strengthened.

**Score:**

4

**Topic Fit:**

3

---

### Official Review · Reviewer_3dww · 2025-09-15
**Review of Universally Converging Representations of Matter Across Scientific Foundation Models**

**Confidence:** 4

**Review:**

This work investigates the Platonic Representation Hypothesis (PRH) in the context of scientific foundation models. The paper is well written and clearly structured, with results that are carefully supported by experimental analysis. To the best of this reviewer’s knowledge, this is the first systematic study of PRH applied to scientific foundation models. Evidence that PRH holds in this domain could have significant ramifications. For example, it could justify training models on cheaper modalities (e.g., simulated or synthetic data) as substitutes for costly experimental modalities, thereby reducing expenses in scientific discovery and accelerating progress.

That said, the results presented are not especially surprising, as one would expect models trained on the same data to be similar, and models to behave based on inductive bias when handling out of distribution data. The novelty lies more in applying PRH to the scientific domain than in the empirical findings themselves. Additionally, the paper would benefit from at least one figure in the main body to provide intuition and improve accessibility for readers.

Overall, this reviewer finds the paper to be a clear, early-stage, and thought-provoking contribution. While the experimental scope is somewhat limited, the framing around PRH is original and has the potential to inspire further research. These results are exciting enough to warrant continued exploration.

**Score:**

4

**Topic Fit:**

3